# Computational Metabolomics Tools Reveal Subarmigerides, Unprecedented Linear Peptides from the Marine Sponge Holobiont *Callyspongia subarmigera*

**DOI:** 10.3390/md20110673

**Published:** 2022-10-27

**Authors:** Andrea Castaldi, Roberta Teta, Germana Esposito, Mehdi A. Beniddir, Nicole J. De Voogd, Sébastien Duperron, Valeria Costantino, Marie-Lise Bourguet-Kondracki

**Affiliations:** 1Molécules de Communication et Adaptation des Microorganismes, UMR 7245 CNRS, Muséum National d’Histoire Naturelle, 57 Rue Cuvier (CP54), 75005 Paris, France; 2The Blue Chemistry Lab Group, Dipartimento di Farmacia, Università degli Studi di Napoli Federico II, Via D. Montesano 49, 80131 Napoli, Italy; 3Équipe “Chimie des Substances Naturelles” BioCIS, CNRS, Université Paris-Saclay, 17 Avenue des Sciences, 91400 Orsay, France; 4Naturalis Biodiversity Center, P.O. Box 9517, 2300 RA Leiden, The Netherlands; 5Institute of Environmental Sciences, Leiden University, Einsteinweg 2, 2333 CC Leiden, The Netherlands

**Keywords:** marine sponge holobiont, *Callyspongia subarmigera*, linear peptide, cyanobacteria, molecular networking, MS/MS

## Abstract

A detailed examination of a unique molecular family, restricted to the *Callyspongia* genus, in a molecular network obtained from an in-house Haplosclerida marine sponge collection (including *Haliclona*, *Callyspongia*, *Xestospongia*, and *Petrosia* species) led to the discovery of subarmigerides, a series of rare linear peptides from *Callyspongia subarmigera*, a genus mainly known for polyacetylenes and lipids. The structure of the sole isolated peptide, subarmigeride A (**1**) was elucidated through extensive 1D and 2D NMR spectroscopy, HRMS/MS, and Marfey’s method to assign its absolute configuration. The putative structures of seven additional linear peptides were proposed by an analysis of their respective MS/MS spectra and a comparison of their fragmentation patterns with the heptapeptide **1**. Surprisingly, several structurally related analogues of subarmigeride A (**1**) occurred in one distinct cluster from the molecular network of the cyanobacteria strains of the Guadeloupe mangroves, suggesting that the true producer of this peptide family might be the microbial sponge-associated community, i.e., the sponge-associated cyanobacteria.

## 1. Introduction

In our previous study [1], a comprehensive metabolomic strategy, integrating ^1^H NMR- and HRMS-based multiblock modelling in conjunction with taxonomically informed molecular networking, was used for the study of 33 Haplosclerida marine sponge samples of three different families (Callyspongiidae, Chalinidae, and Petrosiidae) and four different genera (*Callyspongia* Duchassaing and Michelotti 1864, *Haliclona* Grant 1841, *Petrosia* Vosmaer 1885, and *Xestospongia* de Laubenfels 1932). To inspect the chemical space of the 33 marine sponge extracts, a feature-based molecular network [2] was generated from the layering of their acquired LC-MS/MS and taxonomical data, allowing us to assess the families of compounds that are shared between several species or genera or that are unique to a group. This strategy has proven to be a powerful way to map large dataset collections in order to perform natural product prioritization [3,4,5,6]. As a way to highlight the unique chemistries within this taxonomically homogenous set of samples, the whole molecular network (175 mass features) was mapped at the genus level using a typical color tag (Appendix A). Remarkably, a unique molecular family, containing 56 related nodes, seemed entirely restricted to the *Callyspongia* genus (Appendix A), and the MS/MS fragmentations suggested the presence of linear peptides. This observation stimulated our interest as, so far, the *Callyspongia* genus is only known for its cyclic peptides [7]. For this reason and despite no biological activities being observed in the crude extract, the compound at *m*/*z* 857.4914 belonging to the species *Callyspongia subarmigera* (*Cladochalina*) was isolated in regard to its HPLC profile. Its structure was determined through extensive 1D and 2D NMR spectroscopy, coupled with HRMS/MS data, as an unusual linear heptapeptide that we named subarmigeride A (**1**). The advanced Marfey’s method was used to assign the absolute configuration of the seven amino acids. The putative structures of the seven linear peptides, belonging to the same family, were proposed by an analysis of their respective MS/MS spectra and a comparison of their fragmentation patterns with the heptapeptide subarmigeride A (**1**). Moreover, using the metabolomics tool MASST [8], the annotations matched with the peptide analogues from cyanobacteria, in particular the *Synechocystis* sp. PCC 6803 strain. On the basis of these findings, we have re-examined the cluster of ten linear peptides of the molecular network in cyanobacteria from the mangroves of Guadeloupe [9] (Appendix A) that includes species of the order *Syneccocochales* and *Spirulinales*. A comparison of the MS^2^ spectra revealed these peptides to be analogues of subarmigeride A (**1**) and allowed us to putatively identify seven of them, suggesting a microbial origin for the unusual linear peptide series found in *Callyspongia subarmigera*. The identification of this unusual linear peptide pipeline appeared as a challenge in these *Callyspongia* sponges.

## 2. Results and Discussion

### 2.1. Structure Characterization of Subarmigeride A (**1**) 

Compound **1** was isolated as brown powder. The high-resolution ESI mass spectrum showed [M + H]^+^ ion peaks at *m*/*z* 857.4914, corresponding to the molecular formula C_46_H_64_N_8_O_8_ with 19 unsaturations. The MS/MS fragmentation pattern clearly suggested the peptidic nature of **1**. The fragments at *m*/*z* 716.4146, 630.3298, 533.2763, 472.2919, 342.1805, 239.1387, and 167.1160 originating from the loss of two leucine or isoleucine, two phenylalanine, and three proline residues, in conjunction with the presence of iminium ion peaks characteristics at *m*/*z* 70.0645, 86.0963, and 120.0802, accounted for the presence of three proline, two leucine, and two phenylalanine units. This hypothesis was then confirmed by an NMR analysis (Appendix A), as described below. 

In the ^1^H NMR spectrum, four distinct amide NH signals and seven distinct α-proton signals were present (Table 1). Only four α-protons showed relevant correlation in the TOCSY spectrum with the corresponding amide NH signals (Table 1), confirming the presence of three proline units. The combined analyses of the two-dimensional NMR spectra, i.e., COSY, TOCSY, and HSQC confirmed the presence of two leucine and two phenylalanine residues. An analysis of the HMBC data led to the assignment of the CO signal of each amino acid (except for l-Pro^II^ and l-Leu^II^) through its cross peak with the relevant proton in position 2 and/or 3. The HMBC and NOESY spectra (Table 1) led to the determination of the inter-residue linkages through the correlations between the four amide protons with the carbonyl ^13^C signals of the subsequent amino acids (Leu^I^-NH with Pro^I^-CO, Phe^I^-NH with Phe^II^-CO, Phe^II^-NH with Pro^III^-CO, and Leu^II^-NH with Leu^II^-1′). Regarding the proline units, the HMBC data were used to correlate the protons in position 5 with the α-proton of the subsequent proline (Pro^I^-5 with Pro^I^-2) and the carbonyl ^13^C signals of the subsequent phenylalanine and leucine, respectively (Pro^II^-5 with Phe^I^-1 and Pro^III^-5 with Leu^II^-1). 

In addition, three signals (brs) at δ 7.96 (Leu^II^-1′), 6.97 (Leu^I^-NH_2_ a), and 7.17 (Leu^I^-NH_2_ b) were observed in the proton spectrum. The HMBC correlations between the proton Leu^II^-1′ with the CO-Leu^II^-1, and Leu^I^-NH_2_ a and b with Leu^I^-CO suggested the presence of one terminal formamide and two terminal amide protons, respectively. This agreed with the molecular formula of compound **1** and allowed us to define the planar structure of compound **1** as being the linear heptapeptide NH_2_-Leu^I^-Pro^I^-Pro^II^-Phe^I^-Phe^II-^Pro^III^-Leu^II^-CHO, as depicted in Figure 1, that was named subarmigeride A (**1**). 

As the ΔδC_3_-C_4_ of the three Pro residues were below 8.0 ppm, (4.1, 3.7, and 4.5 ppm, respectively), they were determined to be *trans*. In addition, according to the empirical rule, no NOESY correlation was observed between the Hα of the Pro residues and the Hα of their vicinal amino acid [10]. The structure of the linear heptapeptide **1** was further confirmed by the examination of its MS/MS fragmentation pattern (Table 2, Appendix A). Using the advanced Marfey’s method [11], the configuration of each amino acid was determined to be l-configuration (Appendix A).

### 2.2. Putative Structure of Seven Additional Linear Peptides

A detailed examination of the cluster of the molecular networking revealed that additional related peptides were shared with subarmigeride A (**1**) from the marine sponge *C. subarmigera* (Figure 2). We decided to explore their chemical structure thanks to their respective MS/MS fragmentation spectra (Appendix A). The identification of the putative seven additional peptides was obtained according to their fragmentations, compared with the linear heptapeptide **1** (Table 3). 

### 2.3. Occurrence of the Linear Peptide Subarmigeride A (**1**) in the Previously Studied Cyanobacterium PMC 1052.18 from Guadeloupe 

To further clarify the occurrence of subarmigeride A (**1**), its MS/MS spectrum was queried across all public GNPS datasets [12] using the recently introduced metabolomics tool MASST [7]. Interestingly, among the matched datasets, some analogous compounds occurred in the cyanobacteria genera such as *Synechocystis (Synechocystis* sp. PCC 6803, order *Synechococcales*). 

These results prompted us to re-examine a distinct peptide cluster (Appendix A) in our previous study on cyanobacteria strains from the mangroves of Guadeloupe that showed the occurrence of twelve known peptides from marine sponges in the molecular network annotated with the DEREPLICATOR tool [9]. Satisfyingly, the extracted ion chromatograms of the Guadeloupe cyanobacteria strain PMC 1052.18, corresponding to cyanobacterium gen. nov. 3, sp. nov. 1, revealed the presence of a coincidental feature at *m*/*z* 857.4914 (tolerance < 10 ppm) with a retention time (RT = 5.974 min, Δ = 0.011 min). The MS/MS spectra of both features appeared similar (Appendix A). The strain PMC 1052.18 was isolated from a large benthic bacterial mat. It was initially assigned to a novel genus and species with only limited similarity to other cultured cyanobacterial strains, with a 16S rRNA sequence 90.7% similar to that of the *Synechocystis* sp. PCC 6803 [9]. Recently published, new 16S rRNA sequences available in the GenBank database, as well as further microscopy, allowed us to refine the identification. PMC 1052.18 is closely related to one cyanobacterium assigned to the genus *Spirulina* from soil (strain HSDM2). Indeed, it displays 99% 16SrRNA sequence similarity as well as the helically coiled morphology typical of the genus *Spirulina* (Appendix A).

Despite the similarities in the retention time and fragmentation pattern with the cyanobacteria strain PMC 1052.18, the presence of subarmigeride A (**1**) could not be confirmed. However, a structurally related isomer of this peptide has been detected. This result suggests a potential microbial origin for this unusual linear peptide series found in the marine sponge *Callyspongia subarmigera*, possibly through some associated symbionts, since cyanobacteria have been reported to associate with the genus *Callyspongia*. [13].

## 3. Materials and Methods 

### 3.1. General Experimental Procedures

Mass spectra were recorded on a MAXIS II ETD ultra-high-resolution ESI-QTOF mass spectrometer. NMR spectra were obtained on either a Bruker Avance 400 or 600 spectrometer using standard pulse sequences. Flash chromatography was carried out on Buchi C-615, C-601, C-605 pump system (Rungis, France). Analytical reversed-phase (Luna C18, 250 × 4.6 mm, 5 μm, Phenomenex, Torrance, CA, USA) column was performed with an Agilent Infinity (model 1220 LC), equipped with a photodiode array detector (model 1220 DAD Infinity LC) and the software OpenLab CDS. The data station recorded the wavelengths at 280, 254, and 210 nm. Chromatography columns (CC) were performed using silica gel (200~400 mesh; Merck, Darmstadt, Germany) Sephadex^®^ LH-20 (Amersham Pharmacia, Uppsala, Sweden). The Marfey’s experiment was performed using a Thermo LTQ Orbitrap XL mass spectrometer coupled to a Thermo Ultimate 3000 RS system (Thermo Fisher Scientific Spa, Rodano, Italy), which included solvent reservoir, in-line degasser, ternary pump, column thermostat, and refrigerated autosampler. 

### 3.2. Sponge Material

The sponge sample from this study was identified as *Callyspongia* (*Cladochalina*) *subarmigera*, code name SS18. It belonged to the house sponge extracts collection of the Haplosclerida order, which was collected on South Sulawesi Island (Indonesia). Voucher specimens were deposited at the Naturalis Biodiversity Center.

### 3.3. Sponge Extract Preparation

As previously described [1], the sponge samples (500 g) were cut into small pieces and immediately immersed in MeOH (1 L) after collection. After filtration of an aliquot (20 mL), solvent was evaporated, and 150 mg of each dry extract was mixed to 2 g of C18 and deposited as a powder on a C18 Sep-Pack cartridge (Phenomenex 200 mg/10 mL) to be eluted first with H_2_O (20 mL) in order to eliminate salt and second with MeOH (20 mL) to obtain the desalted extracts. After solvent evaporation, an aliquot of 200 µg was dissolved in 200 µL MeOH for mass analyses.

### 3.4. Isolation and Purification 

The MeOH crude extract (21.03 g) of *Callyspongia subarmigera* was subjected to silica gel flash chromatography and eluted by a gradient mixture of CH_2_Cl_2_/MeOH (0% to 100% MeOH), affording a total of 11 fractions. All of them were analyzed by ultra-high-performance liquid chromatography–tandem mass spectrometry (UPLC-ESI-QToF-MS). Fractions 6–7 showed the presence of subarmigeride A (**1**) (*m*/*z* 857.4914). 

Fractions F6 and F7 were chromatographed over Sephadex^®^ LH20 column using an elution gradient system CH_2_Cl_2_ /MeOH from 0 to 100% MeOH. Therefore, a total of 10 and 8 fractions were obtained, respectively. The subfractions F6-6 and F7-2, analyzed by UPLC-ESI-QToF-MS, showed the presence of subarmigeride A (**1**). 

Subfraction F6-6 was further purified by HPLC with an analytical Gemini C6-Phenyl column (250 × 4.6 mm, 5 μm, Phenomenex) using a gradient system of CH_3_CN/H_2_O/HCOOH (5/95/0.1 to 20/80/0.1 for 60 min, flow rate 1 mL/min, and wavelength 254 nm). 

Subfraction F-7-2 was purified with an analytical reverse phase column (Luna C18, 250 × 4.6 mm, 5 μm, Phenomenex) using a gradient system of CH_3_CN/H_2_O/HCOOH (5/95/0.1 to 20/80/0.1 for 40 min, flow rate 1 mL/min, and wavelength 254 nm). Both subfractions F6-6-4 and F7-2-3 that contained the peptide **1** were further purified by HPLC with a reverse phase column Luna C18 in isocratic conditions, using the mixture H_2_O/ACN 60:40 (0.1% HCOOH in both solvents), for 18 min, at flow rate of 1 mL/min^−1^. Compound **1** was collected from subfractions F6-6-4 and F7-2-3 at 254 nm, with a tR = 11.75 min (0.9 mg and 0.3 mg, respectively). 

Compound (**1**): brown powder; ${\left[\alpha \right]}_{D}^{25}$ + 80 (*c* 0.5, MeOH); ^1^H and ^13^C NMR data: Table 1; and (+)-HRESIMS [M + H]^+^ 857.4914 (calcd. for C_46_H_65_N_8_O_8_^+^, 857.4919). MS/MS spectrum acquired on Bruker Maxis was deposited in the GNPS spectral library under the identifier CCMSLIB00010013071. MS/MS spectrum acquired on Agilent QTOF 6546 was deposited in the GNPS spectral library under the identifier CCMSLIB00010013070.

### 3.5. NMR Data Acquisition and Processing

Proton spectra were acquired at 600 MHz and 298 K on a Bruker Avance III HD spectrometer with a 5 mm reversed TCI cryoprobe. One-dimensional free induction decays (FID) were acquired with a single 90° pulse sequence on 64K data points for 10.0 ppm spectral width, with a 1 s relaxation delay and 256 scan accumulations.

Signal processing was automatically performed in TopSpin software including the Fourier transform with a 0.3 Hz line broadening, baseline correction, and chemical shift calibration (DMSO at *δ*_H_ 2.50 ppm).

2D TOCSY experiment was performed on 2K data points for F2 and 0.5K data points for F1 with 10.0 ppm spectral width in both dimensions, spin lock of 80 ms, and 16 scan accumulations. 2D COSY experiment was performed on 2K data points for F2 and 0.5K data points for F1 with 10.0 ppm spectral width in both dimensions and 16 scan accumulations.

2D NOESY experiment was performed on 2K data points for F2 and 0.5K data points for F1 with 10.0 ppm spectral width in both dimensions, mixing time 500 ms, and 16 scan accumulations.

2D HSQC experiment was performed on 1K data points for F2 and 0.5K data points for F1 with 10.0 ppm spectral width for F2, 190 ppm spectral width for F1, and 16 scan accumulations.

2D HMBC experiment was performed on 2K data points for F2 and 0.5K data points for F1 with 10.0 ppm spectral width for F2, 230 ppm spectral width for F1, and 16 scan accumulations.

### 3.6. LC-MS^2^ Analyses of Extracts 

LC-ESI-HRMS2 analyses were achieved using ultra-high-performance LC system (Ultimate 3000 RSLC, Thermo Scientific, Waltham, MA, USA) coupled to a high-resolution electrospray ionization quadrupole time-of-flight (ESI-Q-TOF) mass spectrometer (MaXis II ETD, Bruker Daltonics, Billerica, MA, USA). An Acclaim RSLC Polar Advantage II column (2.2 µm, 2.1 × 100 mm, Thermo Scientific) was used for LC separation with a flow rate of 0.3 mL/min and a linear gradient from 5% B (A: H_2_O + 0.1% formic acid, B: ACN + 0.08% formic acid) to 100% B in 10 min and then 100% B over 1 min, followed by a decrease to 5% in 1 min for a total runtime of 20 min. The mass range *m*/*z* from 50 to 1300 in positive ion mode was acquired. Injection volume was set at 10 μL. Source parameters were set as follows: nebulizer gas 2.4 bar, dry heater 200 °C, dry gas 8.0 L/min, capillary voltage 3500 V, end plate offset 500 V, and charging voltage 2000 V. For LC-MS/MS, the auto MS/MS mode (collision energy 40.0 eV) was chosen with the same parameters as the MS method. In the first half-minute, calibration solution containing a sodium formate solution was directly injected as an internal reference for calibration. A permanent MS/MS exclusion criteria list was set to prevent oversampling of the internal calibrant. The data were treated with Data Analysis 4.4 (Bruker Daltonics).

Comparison of MS/MS spectra of the feature *m*/*z* 857.4912 (tolerance 10 ppm) was achieved using an Agilent LC–MS system, comprising an Agilent 1260 infinity HPLC coupled to an Agilent 6530 Q-TOF–MS (Agilent Technologies, Massy, France) equipped with an ESI source, operating in positive ion mode. A Sunfire analytical C18 column (150 × 2.1 mm; i.d. 3.5 μm, Waters, Milford, MA, USA) was used, with a flow rate of 250 μL/min and a linear gradient from 5% B (A: H_2_O + 0.1% formic acid, B: ACN) to 100% B in 20 min and then 100% B over 10 min for a total runtime of 30 min. Injection volume was set at 10 μL. Source parameters were set as follows: capillary temperature at 320 °C, source voltage at 3500 V, and sheath gas flow rate at 10 L/ min. The divert valve was set to waste for the first 3 min. MS scans were operated in full-scan mode from *m*/*z* 100 to 1700 (0.1 s scan time) with a mass resolution of 11,000 at *m*/*z* 922. MS1 scan was followed by MS2 scans of the five most intense ions above an absolute threshold of 5000 counts. Selected parent ions were fragmented at a collision energy fixed at 45 eV and an isolation window of 1.3 amu. Calibration solution contained two internal reference masses (purine, C_5_H_4_N_4_, *m*/*z* 121.050873; and HP-921 [hexakis-(1*H*,1*H*,3*H*-tetrafluoropentoxy) phosphazene], C_18_H_18_O_6_N_3_P_3_F_24_, *m*/*z* 922.0098). A permanent MS/MS exclusion list criterion was set to prevent oversampling of the internal calibrant. LC-UV and MS data acquisition and processing were performed using MassHunter Workstation software (Agilent Technologies, Massy, France).

### 3.7. Mass Spectrometry: LC-MS/MS Data Processing

The MS^2^ data file was converted from the .d standard data format to .mzXML format using the MSConvert software, part of the ProteoWizard package [14]. All .mzXML were then imported in MZmine 2 v.53 [15]. The mass detection was performed on exact masses with mass level 1 and centroided masses with mass level 2 by keeping the noise level at 1.2 × 10^3^ at MS^1^ and at 2 × 10^1^ at MS^2^, respectively. The ADAP chromatogram builder was used to build a chromatogram with a minimum group size of scans of 2, a group intensity threshold of 2 × 10^3^, a minimum highest intensity of 2 × 10^3^, and *m*/*z* tolerance of 10 ppm [16]. As it regards chromatogram deconvolution, the local minimum search algorithm was employed with the following settings: chromatographic threshold = 1%, search minimum in RT range (min) = 0.1, minimum relative height = 5%, minimum absolute height = 2 × 10^3^, min ratio of peak top/edge = 1.4, and peak duration range (min) = 0.05–2. MS^2^ scans were paired using a *m*/*z* tolerance range of 0.03 Da and RT tolerance range of 0.15 min. Isotopes were grouped using the isotopic peaks grouper algorithm with a *m*/*z* tolerance of 10 ppm and a RT tolerance of 0.15 min with the lowest peak. [M + Na − H]^+^, [M + K − H]^+^, [M + Mg − 2H]^+^, [M + NH_3_]^+^, [M − Na + NH_4_]^+^, and [M + 1, ^13^C]^+^ adducts were filtered out by setting the maximum relative height at 100%. The resulting peak list was filtered to keep only rows with MS^2^ features. The .mgf and .csv files were generated using the dedicated “Export/Submit to GNPS/FBMN” option.

### 3.8. Mass Spectrometry: Molecular Networking

A molecular network was created using the online FBMN workflow (version release_28.2) on GNPS (https://gnps.ucsd.edu/ProteoSAFe/status.jsp?task=8a40068370b44e21855c1e14647ff23a, accessed on 24 May 2022) (Appendix A). The parent mass tolerance was 0.02 Da, and the MS/MS fragment ion tolerance was 0.02 Da. A network was then created where edges were filtered to have a cosine score above 0.65 and more than 6 matched peaks. Further edges between two nodes were kept in the network if, and only if, each of the nodes appeared in the respective top 10 most similar nodes of each other. The spectra in the network were then searched against GNPS spectral libraries. All matches kept between network spectra and library spectra were required to have a score above 0.65 and at least 6 matched peaks. The molecular networking data were analyzed and visualized using Cytoscape (ver. 3.9.1) [17].

### 3.9. Advanced Marfey’s Analysis 

According to previously described experiment [10], compound **1** (200 µg) was hydrolyzed with 6 N HCl/AcOH (1:1) at 120 °C for 12 h. The residual HCl fumes were removed under N_2_ stream. The hydrolysate of 1 was dissolved in TEA/acetone (2:3, 100 μL), and the solution was treated with 100 μL of 1% 1-fluoro-2,4-dinitrophenyl-5-d-alaninamide (d-FDAA) in ACN/acetone (1:2). The vial was heated at 50 °C for 2 h. The mixture was dried, and the resulting d-FDAA derivatives of Leu, Phe, and Pro were redissolved in MeOH (100 μL) for subsequent analysis. Authentic standards of l-Pro, l-Phe, and l-Leu were treated with l-FDAA and d-FDAA as described above and yielded the l-FDAA and d-FDAA standards. Marfey’s derivatives of 1 were analyzed using HPLC-ESI-HRMS, and their retention times were compared with those from the authentic standards derivatives. A Kinetex C18 (Phenomenex) 150 × 2.1 mm, 5 μm column. The gradient conditions were set as follows: 35 min prerun with 5% ACN, 5% ACN 3 min, 5%→50% ACN over 30 min, 50% ACN 1 min, 50%→90% ACN 1 min, and 90% ACN 6. Mass spectra were acquired in positive ion detection mode, and the data were analyzed using the suite of programs XCalibur. 

### 3.10. MASST Analysis

A single spectrum search was completed using the online workflow MASST (workflow version release_29) on the GNPS website (http://gnps.ucsd.edu, accessed on 2 September 2022) across all GNPS datasets. The data were filtered by removing all MS/MS fragment ions within +/− 17 Da of the precursor *m*/*z*. MS/MS spectra were window-filtered by choosing only the top 6 fragment ions in the +/− 50 Da window throughout the spectrum. The precursor ion mass tolerance was set to 2.0 Da and a MS/MS fragment ion tolerance of 0.5 Da. The library spectra were filtered in the same manner as the input data. All matches kept between input spectra and library spectra were required to have a score above 0.2 and at least 3 matched peaks. The job is accessible here: https://gnps.ucsd.edu/ProteoSAFe/status.jsp?task=c118cea33a4443478caf118c82d4ec98, accessed on 2 September 2022.

### 3.11. Mass Spectrometry: Comparison of LC-MS/MS Data of the Cyanobacterial Strain PMC 1052.18 and the Marine Sponge Callyspongia subarmigera

UPLC-ESI-HRMS2 analyses were achieved by coupling the UPLC system to a hybrid quadrupole time-of-flight mass spectrometer Agilent 6546 (Agilent Technologies, Massy, France) equipped with an ESI source, operating in both positive and negative ion mode. A BEH Acquity C18 UPLC column (2.1 × 150 mm; i.d. 1.8 µm, Waters) was used, with a flow rate of 0.5 mL/min and a linear gradient from 5% B (A: H_2_O + 0.1% formic acid, B: Acetonitrile + 0.1% formic acid) to 100% B over 15 min. Source parameters were set as follows: capillary temperature at 320 °C, source voltage at 3500 V, and sheath gas flow rate at 11 L/min. The divert valve was set to waste for the first 3 min. MS and MS2 scans were operated in full-scan mode from *m*/*z* 70 to 1200 (0.1 s scan time) with a mass resolution of 67,000 at *m*/*z* 922. A MS1 scan was followed by MS2 scans of the five most intense ions above an absolute threshold of 3000 counts. Selected parent ions were fragmented at a collision energy fixed at 45 eV and an isolation window of 1.3 amu. In the positive ion mode, purine C_5_H_4_N_4_ [M + H]^+^ ion (*m*/*z* 121.050873) and the hexakis (1*H*,1*H*,3*H*-tetrafluoropropoxy)-phosphazene C_18_H_18_F_24_N_3_O_6_P_3_ [M + H]^+^ ion (*m*/*z* 922.009798) were used as internal lock masses. A permanent MS/MS exclusion list criterion was set to prevent oversampling of the internal calibrant. LC-UV and MS data acquisition and processing were performed using MassHunter^®^ Workstation software (Agilent Technologies, Massy, France).

## 4. Conclusions

A detailed examination of the whole molecular network of an in-house collection of Haplosclerida marine sponges led to the selection and deep analysis of the crude extract of the marine sponge *Callyspongia subarmigera* (*Cladochalina*) and to the first discovery of one unusual linear peptide that we named subarmigeride A (**1**). So far, only cyclic peptides have been reported in the literature within the genus *Callyspongia* (261 described species) marine sponges, which are mainly known for their polyacetylenes and lipids [7]. Furthermore, the putative seven additional peptides, revealed by fragmentation pattern analyses, contribute to the knowledge of the chemical diversity of marine sponges of the genus *Callyspongia*. Although the biological activities of *Callyspongia* marine sponges have been previously reported, the evaluation of the crude extract of *C. subarmigera* at 10 µg/mL against the human lung adenocarcinoma (A549), colorectal carcinoma (HCT116), and leukemia (HL60) cell lines showed no cytotoxic activity. Furthermore, no environmental activities, including antibiofilm activity against *Pseudomonas aeruginosa* and antihelminthic activity, have been detected. Consequently, the role of the isolated peptide subarmigeride A (**1**) and its congeners within the marine sponge *Callyspongia subarmigera* remains to be elucidated. Furthermore, the purification of subarmigeride A and its analogues are in progress in order to obtain pure linear peptides that could be evaluated for biological and environmental activities. Interestingly, the occurrence of structurally related analogues produced by cyanobacteria was revealed using MASST. These results suggest that the reported linear peptides might originate from cyanobacteria, which are well-known producers of linear peptides [18]. The present study confirms the prolific source of unusual molecules produced by the marine sponge holobiont as well as the interest of sharing well-informed omics datasets in the context of microbiome research. 

## Figures and Tables

**Figure 1 marinedrugs-20-00673-f001:**
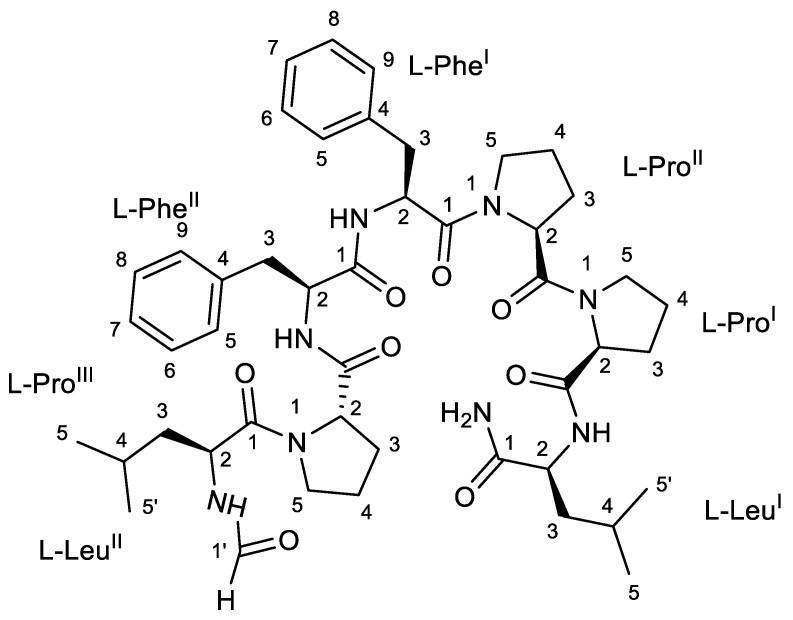
Structure of the linear peptide subarmigeride A (**1**).

**Figure 2 marinedrugs-20-00673-f002:**
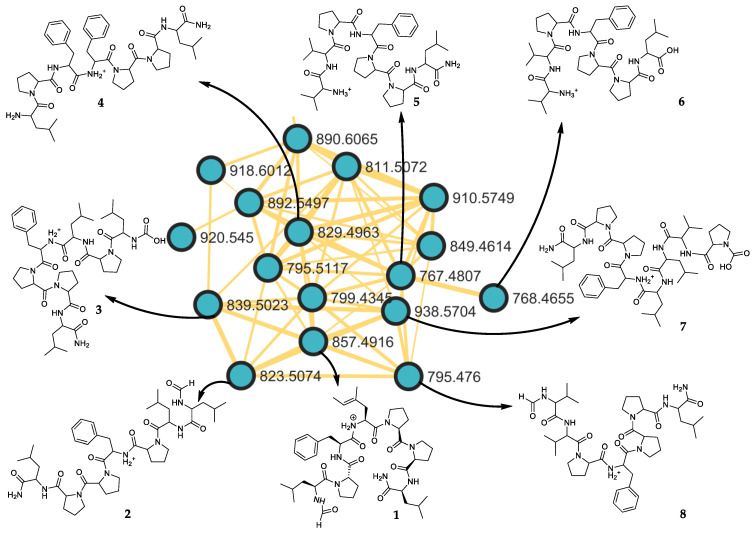
Zoom on the cluster of peptides present in the marine sponge *Callyspongia subarmigera*.

**Table 1 marinedrugs-20-00673-t001:** ^1^H, ^13^C, HMBC, and NOESY NMR data of subarmigeride A (**1**) at 600 MHz (DMSO-*d6*).

Unit	Pos.	δ_C,_ Mult.		δ_H_ (*J* in Hz)	HMBC	NOESY ^a^
l-Leu^I^	NH	-		7.71, d (8.22)	Leu^I^-1, Pro^I^-CO	Pro^I^-1, Pro^I^-2-a, Pro^I^-2-b, Pro^I^-3-a, Pro^I^-3-b, Pro^I^-4-a
	1	174.1, C		-	-	-
	2	50.7, CH		4.16, ddd (6.72, 8.22, 14.93)	Leu^I^-CO, Leu^I^-2, Leu^I^-3, Pro^I^-CO	Pro^I^-1
	3	40.7, CH_2_		1,47, m	Leu^I^-1, Leu^I^-3, Leu^I^-4, Leu^I^-4′	-
	4	24.2, CH		1.61, m	Leu^I^-2, Leu^I^-4, Leu^I^-4′	Pro^I^-2-a, Pro^I^-2-b
	5	23.1, CH_3_		0.87, d (6.59)	Leu^I^-2, Leu^I^- 3, Leu^I^-4′	-
	5′	21.5, CH_3_		0.83, d (6.59)	Leu^I^-2, Leu^I^- 3, Leu^I^-4	-
	NH_2_	-	a. b.	6.97, brs 7.17 brs	Leu^I^-CO, Leu^I^-1	- -
l-Pro^I^	1	171.2, C		-	-	-
	2	59.7, CH		4.31 dd (4.42, 8.28)	Pro^I^-3	Leu^I^-NH
	3	28.7, CH_2_	a. b.	1.80 m 2.02 m	Pro^I^-CO	Leu^I^-NH, Leu^I^-3 Leu^I^-NH, Leu^I^-4
	4	24.6, CH_2_	a. b.	1.86 m 1.88 m	- -	Leu^I^-NH Leu^I^-NH
	5	46.7, CH_2_	a. b.	3.52 m 3.67 m	Pro^I^-2	Leu^I^-NH, Pro^II^-1, Pro^II^-2-b Pro^II^-1, Pro^II^-2-b
l-Pro^II^	1	170.1, C *		-	-	-
	2	57.8, CH		4.53, dd (4.45, 8.29)	-	Phe^I^-1, Pro^I^-3-a, Pro^I^-3-b
	3	28.0, CH_2_	a. b.	1.84 m 2.09 m	- -	- Pro^I^-4-a, Pro^I^-4-b
	4	24.3, CH_2_	a. b.	1.81 m 1.92 m	- -	- Phe^I^-1
	5	46.6, CH_2_	a. b.	3.37 m 3.57 m	- -	Phe^I^-1, Phe^I^-2-b Phe^I^-1
l-Phe^I^	NH	-		8.17, d (8.20)	Phe^II^-CO	Phe^II^-1, Phe^II^-NH, Leu^II^-NH
	1	168.9, C		-	-	-
	2	51.8, CH		4.64, ddd (5.05, 8.20, 13.26)	Phe^I^-CO	Pro^II^-1, Pro^II^-3-b, Phe^II^-1, Pro^II^-4-a, Pro^II^-4-b
	3	36.7, CH_2_	a. b	2.76 dd (1.90, 8.20) 2.95 dd (4.87, 14.08)	Phe^I^-CO, Phe^I^-1, Phe^I^-1, Phe^I^-2/6, Phe^II^-CO	- Pro^II^-4-a
	4	137.4, C		-	-	-
	5/9	129.2, CH		7.25 ^b^	Phe^I^-2, Phe^I^-4, Phe^I^-6	-
	6/8	128.1, CH		7.23 ^b^	Phe^I^-1, Phe^I^-5	-
	7	126.3, CH		7.17 ^b^	Phe^I^-2/6	-
l-Phe^II^	NH	-		7.74, d (8.05)	Pro^III^-CO	Phe^I^-1, Pro^III^-1, Pro^III^-2-a, Pro^III^-2-b, Pro^III^-3-b
	1	170.3, C		-	-	-
	2	53.5, CH		4.41, ddd (5.37, 8.05, 13.42)	Phe^II^-CO, Phe^II^-5-2	Phe^I^-1, Phe^I^-NH, Pro^III^-1, Leu^II^-NH
	3	37.2, CH_2_	a. b.	2.78 dd (2.17, 10.02), 2.92 dd (5.42, 14.34)	Phe^II^-CO, Phe^II^-1, Phe^II^-1, Phe^I^-2/6	- -
	4	137.4, C		-	-	-
	5/9	129.2, CH		7.15 ^b^	Phe^II^-2, Phe^II^-4, Phe^II^-6	-
	6/8	127.9, CH		7.20 ^b^	Phe^II^-1, Phe^II^-5	-
	7	126.1, CH		7.16 ^b^	Phe^II^-2/6	-
l-Pro^III^	1	170.9, C		-	-	-
	2	59.2, CH		4.30, dd (4.45, 8.29)	-	Phe^II^-NH, Phe^II^-1, Leu^II^-1
	3	28.7, CH_2_	a. b.	1.73 m 1.91 m	- -	Phe^II^-NH Phe^II^-NH
	4	24.2, CH_2_	a. b	1.76 m 1.81 m	- -	- Phe^II^-NH
	5	46.6, CH_2_	a. b.	3.42 m 3.60 m	- -	Leu^II^-1, Leu^II^-2-a, Leu^II^-2-b Leu^II^-1, Leu^II^-7-2-a, Leu^II^-2-b
l-Leu^II^	NH	-		8.29 dd (1.31, 8.34)	Leu^II^-HCO	Phe^II^-1, Phe^I^-NH
	1	169.8, C *		-	-	-
	2	47.1, CH		4.58 ddd (4.31, 8.34, 12.09)	Pro^III^-CO	Pro^III^-1, Leu^II^-3-a, Pro^III^-3-b
	3	40.2, CH_2_	a. b	1.39 m 1.38 m	Pro^III^-3, Pro^III^-4′, Pro^III^-4′	Pro^III^-4-a, Pro^III^-4-b Pro^III^-4-a, Pro^III^-4-b
	4	24.0, CH		1.57, m	-	-
	5	21.3, CH_3_		0.85, d (1.68)	Pro^III^-2, Pro^III^-3, Pro^II^^I^-4′	-
	5′	23.1, CH_3_		0.84, d (1.89)	Pro^III^-2, Pro^III^-3, Pro^III^-4	-
	HCO	160.7, CH		7.96 brs	Leu^II^-1	-

^a^ Sequential NOEs. ^b^ Overlapped signals prevent determination of constant couplings. * May be interchanged.

**Table 2 marinedrugs-20-00673-t002:** Product ion spectra data for subarmigeride A (**1**) (*m*/*z* 857.4914 [M + H]^+^).

Product Ion Assignment	(*m*/*z*)	Error, pm	Molecular Formula
CHO-Leu-Pro-Phe-Phe-Pro-Pro-Leu-NH_2_ + H^+^	857.4914	0.7	C_46_H_65_N_8_O_8_
Pro-Phe-Phe-Pro-Pro-Leu-NH_2_ + H^+^	716.4146	−2.2	C_39_H_54_N_7_O_6_
Pro-Phe-Phe-Pro-Pro-Leu + H^+^	699.3862	0.4	C_39_H_51_N_6_O_6_
Pro-Phe-Phe-Pro-Pro-Leu + H^+^	671.3931	−2.3	C_38_H_51_N_6_O_5_
Phe-Phe-Pro-Pro-Leu + H^+^	630.3298	−1.9	C_35_H_44_N_5_O_6_
Phe-Phe-Pro-Pro + H^+^	533.2763	−0.8	C_30_H_37_N_4_O_5_
CHO-Leu-Pro-Phe-Phe + H^+^	505.2813	−0.8	C_29_H_37_N_4_O_4_
Phe-Phe-Pro-Pro-NH_2_ + H^+^	489.2506	−1.9	C_28_H_33_N_4_O_4_
Phe-Pro-Pro-Leu-NH_2_ + H^+^	472.2919	0	C_25_H_38_N_5_O_4_
Phe-Pro-Pro-Leu-NH_2_ + H^+^	455.2659	−1.4	C_25_H_35_N_4_O_4_
Phe-Phe-Pro + H^+^	392.1971	−0.6	C_23_H_26_N_3_O_3_
CHO-Leu-Pro-Phe + H^+^	386.2075	−0.2	C_21_H_28_N_3_O_4_
Pro fragment-Phe-Phe + H^+^	375.1700	0.9	C_23_H_23_N_2_O_3_
CHO-Leu-Pro-Phe + H^+^	358.2129	−1	C_20_H_28_N_3_O_3_
Phe-Pro-Pro + H^+^	342.1805	2.2	C_19_H_24_N_3_O_3_
Pro-Pro-Leu-NH_2_ + H^+^	325.2232	0.6	C_16_H_29_N_4_O_3_
Pro-Pro-Leu + H^+^	308.1966	0.8	C_16_H_26_N_3_O_3_
Phe-Pro-NH_2_ + H^+^	245.1282	1.2	C_14_H_17_N_2_O_2_
Pro-Leu + H^+^	239.1387	1.4	C_12_H_19_N_2_O_3_
Pro-Leu-NH_2_ + H^+^	228.1702	1.8	C_11_H_22_N_3_O_2_
Pro-Leu + H^+^	217.1332	1.4	C_13_H_17_N_2_O
Pro-Leu + H^+^	211.1437	1.7	C_11_H_19_N_2_O_2_
Pro-Pro + H^+^	195.1124	1.8	C_10_H_15_N_2_O_2_
Pro-Leu + H^+^	183.1485	3.5	C_10_H_19_N_2_O
Pro-Pro + H^+^	167.1160	11.2	C_9_H_15_N_2_O
Phe immonium fragment + H^+^	120.080173	5	C_8_H_10_N
Leu immonium fragment + H^+^	86.096342	1	C_5_H_12_N
Pro immonium fragment + H^+^	70.064503	8.9	C_4_H_8_N

**Table 3 marinedrugs-20-00673-t003:** Putative structures of the linear peptides **2** to **8** from *Callyspongia*
*subarmigera*.

Subarmigeride at *m*/*z* *	Molecular Formula [M + H]^+^	aa-1	aa-2	aa-3	aa-4	aa-5	aa-6	aa-7	aa-8
B (**2**) at 823.5078	C_43_H_67_N_8_O_8_^+^	NH_2_-Leu/Ile	Pro	Pro	Phe	Pro	Leu/ Ile	Leu/Ile-CHO	
C (**3**) at 839.5024	C_43_H_67_N_8_O_9_^+^	NH_2_-Leu/Ile	Pro	Pro	Phe	Leu/ Ile	Pro	Leu/Ile-COOH	
D (**4**) at 829.4962	C_45_H_65_N_8_O_7_^+^	NH_2_-Leu/Ile	Pro	Pro	Phe	Phe	Pro	Leu/Ile	
E (**5**) at 767.4807	C_40_H_63_N_8_O_7_^+^	NH_2_-Leu/Ile	Pro	Pro	Phe	Pro	Val	Val	
F (**6**) at 768.4650	C_40_H_63_N_7_O_8_^+^	Leu/Ile	Pro	Pro	Phe	Pro	Val	Val	
G (**7**) at 938.5698	C_48_H_76_N_9_O_10_^+^	NH_2_-Leu/Ile	Pro	Pro	Phe	Leu/ Ile	Leu/ Ile	Val	Pro-COOH
H (**8**) at 795.4763	C_41_H_63_N_8_O_8_^+^	NH_2_-Leu/Ile	Pro	Pro	Phe	Pro	Val	Val-CHO	

*** From Maxis II EDT Q-Tof-MS (exact mass of MS1).

## Data Availability

Not applicable.

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
