# Peer review of "Computational Metabolomics Tools Reveal Subarmigerides, Unprecedented Linear Peptides from the Marine Sponge Holobiont Callyspongia subarmigera"

_marinedrugs, 2022, doi:10.3390/md20110673_

Round 1
Reviewer 1 Report
This article introduces the discovery of subarmigerides, which is an interesting study. However, the overall logic of this paper has yet to be perfected, and the purpose of this research was not well explained. Some specific comments were listed as follows:
1) Is subarmiride A (1) first introduced in this paper? The authors purified this peptide through such complicated steps, so I think the biological activity and potential application of this heptapeptide should be discussed in the introduction section.
2) Except for subarmigeride A (1), how the other seven peptides were selected and why these seven peptides were selected and discussed? I could not get the point from the paper.
3) The method in 3.3 should be written in more detail, such as the main chromatographic and mass spectrometry parameters.
4) Are the relevant parameter settings in MASST analysis reasonable? Is there any relevant literature to support it? Will it result in a false match?
5) The method section lacks the descriptions for NMR experiment.
6) The authors demonstrated that the target peptide may come from cyanobacteria through molecular networking analysis, but it was not verified in the end. Further verification and interpretation of this conclusion is recommended.
Author Response
Dear Reviewer, We are very grateful to the Reviewers for thoroughly checking our manuscript and their valuable suggestions and remarks. Please, find a revised manuscript titled “MD-1960037-revised version”, in which all revisions have been written in red color. In addition, for the revision of the manuscript, we run again the analysis of the strain PMC 1052.18, that displayed 90.7% sequence similarity with the cyanobacterial genus Synechocystis (Synechococcales). We have also performed additional microscopy-based affiliation, that led to a Spirulina based on morphology and 16S rRNA sequences in GENBANK. These results were not available when PMC 1052.18 was initially published in 2020. This new information was added to the manuscript where necessary : Line 63: “…that includes species of the order Syneccocochales and Spirulinales.” Line 125: “Occurrence of the Linear Peptide Subarmigeride A (1) in the Previously Studied Cyanobacterium PMC 1052.18 from Guadeloupe” Lines 140-147: “It was initially assigned to a novel genus and species with only limited similarity to other cultured cyanobacterial strains, with 16S rRNA sequence 90.7% similar to that of Synechocystis sp. PCC 6803 [9]. Recently published new 16S rRNA sequences available in the Genbank database as well as further microscopy allowed us to refine identification. PMC 1052.18 is closely related to one cyanobacterium assigned to genus Spirulina from soil (strain HSDM2). Indeed, it displays 99% 16SrRNA sequence similarity as well as the helically coiled morphology typical of genus Spirulina (Figure S20).” Lines 391-393: “Comparison of MS/MS spectra of the feature m/z 857.4920 at 6.036 min in the cyanobacterial strain PMC 1052.18 (Spirulina sp.) from a mangrove in Guadeloupe” Furthermore, we have added an additional supplementary figure (Figure S20) showing the morphology of the strain PMC 1052.18. In addition, the Museum code name of the marine sponge Callyspongia subarmigera (MNHN-CX-0108) was added in line 414. The respond to the reviewer’s new comments, written in blue color, are as follows: Reviewer 1 : The authors thank Reviewer 1 for its efficient remarks and comments. |
Comments and Suggestions for Authors
This article introduces the discovery of subarmigerides, which is an interesting study. However, the overall logic of this paper has yet to be perfected, and the purpose of this research was not well explained. Some specific comments were listed as follows:
1) Is subarmiride A (1) first introduced in this paper? The authors purified this peptide through such complicated steps, so I think the biological activity and potential application of this heptapeptide should be discussed in the introduction section.
In order to improve the introduction section for facilitating the reading, the following paragraph (lines 46-54) was added: “Remarkably, a unique molecular family, containing 56 related nodes, seemed entirely restricted to the Callyspongia genus (Figure S1B), which MS/MS fragmentations suggested the presence of linear peptides. This observation stimulated our interest as so far, Callyspongia genus is only known for its cyclic peptides [7]. For this reason and despite no biological activities was observed in the crude extract, the compound at m/z 857.4914 belonging to the species Callyspongia subarmigera (Cladochalina) was isolated in regards to its hplc profile. Its structure was determined through extensive 1D and 2D NMR spectroscopy, coupled with HRMS/MS data as an unusual linear heptapeptide, that we named subarmigeride A (1).”
2) Except for subarmigeride A (1), how the other seven peptides were selected and why these seven peptides were selected and discussed? I could not get the point from the paper.
Detailed examination of the molecular networking obtained through the LC-MS/MS analysis of cyanobacterial strains extracts from Guadeloupe’s mangroves, revealed the presence of 10 linear peptides, structurally analogues of those found in the cluster of peptides presents in the marine sponge Callyspongia subarmigera. We have selected seven out of them from Callyspongia subarmigera that could be considered as analogues of the peptides find in molecular networking of cyanobacteria from Guadeloupe’s mangroves (Figure S2) on the basis of their similar masses and fragmentation patterns.
For clarification, these information have been specified in the introduction section (Lines 61-67)“.
3) The method in 3.3 should be written in more detail, such as the main chromatographic and mass spectrometry parameters.
The method in 3.3 is only focused on the preparation of the crude extract of the marine sponge Callyspongia subarmigera. All the details regarding purification including chromatographic steps are detailed in the following 3.4 paragraph. Furthermore, all mass spectrometry parameters were added (Lines 232-248) and are now detailed in paragraph 3.6 titled” LC-MS2 analyzes of extracts”.
4) Are the relevant parameter settings in MASST analysis reasonable? Is there any relevant literature to support it? Will it result in a false match?
We thank the referee for this comment. The parameters that were used here are the default ones recommended on the GNPS website. We used tolerant parameters on purpose in order to match a maximum of datasets in a variable dereplication mode. Indeed, since the purpose beneath the use of MASST is to explore public metabolomics datasets (acquired on different instruments with different accuracies), we were not expecting exactly the same compound but analogous ones. As a reminder, MASST here helped us to realize that cyanobacteria datasets may contain analogous compounds structurally related to the new described family of peptides (subarmigerides). Afterwards, we explored the LC-MS/MS data of the cyanobacteria extract that we had in order to evaluate what MASST suggested to us.
5) The method section lacks the descriptions for NMR experiment.
As recommended, a paragraph titled “3.5 NMR Data Acquisition and Processing“ was added (Lines 207-228).
6) The authors demonstrated that the target peptide may come from cyanobacteria through molecular networking analysis, but it was not verified in the end. Further verification and interpretation of this conclusion is recommended.
We understand the concerns of the referee. Indeed, finding analogous compounds of our peptide in cyanobacteria genera such as Spirulina sp. was interesting. Since we were not able to isolate the matched compound in the cyanobacteria extract, we provided a putative annotation based on analytical read out (HRMS, MS/MS, Rt) as disclosed in the SI (Figure S19). Supporting our results are reports since 1993 (Chem Rev, 1993, doi: 10.1021/cr00021a007), that described sponges as providing housing for macro- and micro-organisms, including cyanobacteria, algae, and dinoflagellates. Recently, we found that
a common biosynthetic machinery is present in free-living cyanobacterium Trichodesmium sp. and the holobiome of the sponge Smenospongia aurea (Org. Chem. Frontiers., 2019, doi: :10.1039/c9qo00074g). So, data reported in the present paper are an additional proof of the microbial origin of peptides. Further studies will be done to add more details to this
finding.
We hope meet with your approval.
Looking forward to your reply,
Sincerely yours,
Dr. Marie-lise Bourguet-Kondracki

Reviewer 2 Report
In this manuscript, the authors describe their results on a family of molecules unique to the genus Callyspongia. In the sponge Callyspongia subarmigera, they identified a group of linear peptides, subarmigerides. The structure, including the absolute configuration, of subarmigeride A, consisting of seven amino acids, has been elucidated and is presented in this manuscript. In addition, the putative structures of seven analogues of subarmigeride A are proposed based on the analysis of MS/MS spectra and fragmentation patterns. Cluster analysis of the molecular network of these molecules and peptides from cyanobacteria strains from the mangroves of Guadeloupe islands led to the conclusion that the sponge subarmigerides in C. subarmigera could originate from these sponge-associated microbes.
This is an interesting study that presents new insights into a unique group of peptides only found in Callyspongia. The data are carefully presented and support the conclusions. This manuscript is in the scope of Marine Drugs and is surely of interest to the readership of the journal.
Unfortunately, no bioactivities have yet been found for this unusual group of linear heptapeptide molecules. The authors mention negative results in terms of cytotoxicity using different human cell lines (concentration tested 10 μg/mL) and antibiofilm activity (Pseudomonas aeruginosa) and anti-helminthic activity. Perhaps the authors have an idea about possible biological function and activity of these molecules. You should add a sentence or two to this question and their future strategy in studying this interesting group of peptides.
I strongly propose to accept this manuscript after minor revision.
The required revision mainly relates to the English and some typos, some of which are mentioned below.
Line 81: “L-ProI” or “1-fluoro-2,4-dinitrophenyl-5-d-alaninamide” etc. Please use l (small capital letter) and d (small capital letter) instead of capital letters or small type letters.
Line 83: "through" instead of "trough"
Line 92: “terminal formamide” instead of “formamide terminal”
Line 107: “…the configuration of each amino acid was determined to be …” instead of “…the configuration of each amino acid was determined as being…”
Line 114-115: “Identification of … was obtained …” instead of “Identification of … were obtained”
Line 123: “To further clarify the occurrence of subarmigeride A …” instead of “To further question the occurrence of subarmigeride A …”
Line 130: “marine sponges” instead of “marine Sponges”
Line 298-301: Please rephrase this sentence, for example: “… led to the selection and analysis of the crude extract of …, leading to the discovery of ….”
Line 306-307: “Although biological activities of Callyspongia marine sponges have been previously reported, …” instead of “Although biological activities have been previously reported from Callyspongia marine sponges, ..”
Line 307: Please add “adenocarcinoma”, i.e. “human lung adenocarcinoma (A549) .. cell line “ instead of “human lung (A549)… cell line”
Line 310-312: “Consequently, the role of … remains to...” Instead of “Consequently, the role of … remain to …“
Lines 313-314: “Interestingly, the occurrence of … produced by cyanobacteria was revealed using MASST.” Instead of “Interestingly, the occurrence of … produced by cyanobacteria, were revealed with MASST tool.” (“tool” is already contained in the abbreviation MASST - “Mass Spectrometry Search Tool)
Lines 314-316: “These results suggest that the reported linear peptides might originate from cyanobacteria, which are well known producers of linear peptides” instead of “These results suggested that the reported linear peptides might have a cyanobacterial origin, which are well known as producers of linear peptides”
Minor comment:
In the subheadings, capital letters are sometimes used at the beginning of all words, sometimes not. Please make it consistent.
This manuscript should be accepted after minor revision.
Author Response
Dear Reviewer,
We are very grateful to the Reviewers for thoroughly checking our manuscript and their valuable suggestions and remarks.
Please, find a revised manuscript titled “MD-1960037-revised version”, in which all revisions have been written in red color.
In addition, for the revision of the manuscript, we run again the analysis of the strain PMC 1052.18, that displayed 90.7% sequence similarity with the cyanobacterial genus Synechocystis (Synechococcales). We have also performed additional microscopy-based affiliation, that led to a Spirulina based on morphology and 16S rRNA sequences in GENBANK. These results were not available when PMC 1052.18 was initially published in 2020.
This new information was added to the manuscript where necessary :
Line 63: “…that includes species of the order Syneccocochales and Spirulinales.”
Line 125: “Occurrence of the Linear Peptide Subarmigeride A (1) in the Previously Studied Cyanobacterium PMC 1052.18 from Guadeloupe”
Lines 140-147: “It was initially assigned to a novel genus and species with only limited similarity to other cultured cyanobacterial strains, with 16S rRNA sequence 90.7% similar to that of Synechocystis sp. PCC 6803 [9]. Recently published new 16S rRNA sequences available in the Genbank database as well as further microscopy allowed us to refine identification. PMC 1052.18 is closely related to one cyanobacterium assigned to genus Spirulina from soil (strain HSDM2). Indeed, it displays 99% 16SrRNA sequence similarity as well as the helically coiled morphology typical of genus Spirulina (Figure S20).”
Lines 391-393: “Comparison of MS/MS spectra of the feature m/z 857.4920 at 6.036 min in the cyanobacterial strain PMC 1052.18 (Spirulina sp.) from a mangrove in Guadeloupe”
Furthermore, we have added an additional supplementary figure (Figure S20) showing the morphology of the strain PMC 1052.18.
In addition, the Museum code name of the marine sponge Callyspongia subarmigera (MNHN-CX-0108) was added in line 414.
Reviewer 2 :
The authors thank Reviewer 2 for its efficient remarks and comments.
Comments and Suggestions for Authors
In this manuscript, the authors describe their results on a family of molecules unique to the genus Callyspongia. In the sponge Callyspongia subarmigera, they identified a group of linear peptides, subarmigerides. The structure, including the absolute configuration, of subarmigeride A, consisting of seven amino acids, has been elucidated and is presented in this manuscript. In addition, the putative structures of seven analogues of subarmigeride A are proposed based on the analysis of MS/MS spectra and fragmentation patterns. Cluster analysis of the molecular network of these molecules and peptides from cyanobacteria strains from the mangroves of Guadeloupe islands led to the conclusion that the sponge subarmigerides in C. subarmigera could originate from these sponge-associated microbes.
This is an interesting study that presents new insights into a unique group of peptides only found in Callyspongia. The data are carefully presented and support the conclusions. This manuscript is in the scope of Marine Drugs and is surely of interest to the readership of the journal.
Unfortunately, no bioactivities have yet been found for this unusual group of linear heptapeptide molecules. The authors mention negative results in terms of cytotoxicity using different human cell lines (concentration tested 10 μg/mL) and antibiofilm activity (Pseudomonas aeruginosa) and anti-helminthic activity. Perhaps the authors have an idea about possible biological function and activity of these molecules. You should add a sentence or two to this question and their future strategy in studying this interesting group of peptides.
I strongly propose to accept this manuscript after minor revision.
The required revision mainly relates to the English and some typos, some of which are mentioned below.
Line 81: “L-ProI” or “1-fluoro-2,4-dinitrophenyl-5-d-alaninamide” etc. Please use l (small capital letter) and d (small capital letter) instead of capital letters or small type letters.
Line 83: "through" instead of "trough"
Line 92: “terminal formamide” instead of “formamide terminal”
Line 107: “…the configuration of each amino acid was determined to be …” instead of “…the configuration of each amino acid was determined as being…”
Line 114-115: “Identification of … was obtained …” instead of “Identification of … were obtained”
Line 123: “To further clarify the occurrence of subarmigeride A …” instead of “To further question the occurrence of subarmigeride A …”
Line 130: “marine sponges” instead of “marine Sponges”
Line 298-301: Please rephrase this sentence, for example: “… led to the selection and analysis of the crude extract of …, leading to the discovery of ….”
Line 306-307: “Although biological activities of Callyspongia marine sponges have been previously reported, …” instead of “Although biological activities have been previously reported from Callyspongia marine sponges, ..”
Line 307: Please add “adenocarcinoma”, i.e. “human lung adenocarcinoma (A549) .. cell line “ instead of “human lung (A549)… cell line”
Line 310-312: “Consequently, the role of … remains to...” Instead of “Consequently, the role of … remain to …“
Lines 313-314: “Interestingly, the occurrence of … produced by cyanobacteria was revealed using MASST.” Instead of “Interestingly, the occurrence of … produced by cyanobacteria, were revealed with MASST tool.” (“tool” is already contained in the abbreviation MASST - “Mass Spectrometry Search Tool)
Lines 314-316: “These results suggest that the reported linear peptides might originate from cyanobacteria, which are well known producers of linear peptides” instead of “These results suggested that the reported linear peptides might have a cyanobacterial origin, which are well known as producers of linear peptides”
Minor comment:
In the subheadings, capital letters are sometimes used at the beginning of all words, sometimes not. Please make it consistent.
This manuscript should be accepted after minor revision.
We thank Reviewer 2 for his efficient remarks and comments. All the suggested corrections have been done.
In order to improve the introduction section for facilitating the reading, the following paragraph (lines 46-54) was added: “Remarkably, a unique molecular family, containing 56 related nodes, seemed entirely restricted to the Callyspongia genus (Figure S1B), which MS/MS fragmentations suggested the presence of linear peptides. This observation stimulated our interest as so far, Callyspongia genus is only known for its cyclic peptides [7]. For this reason and despite no biological activities was observed in the crude extract, the compound at m/z 857.4914 belonging to the species Callyspongia subarmigera (Cladochalina) was isolated in regards to its hplc profile. Its structure was determined through extensive 1D and 2D NMR spectroscopy, coupled with HRMS/MS data as an unusual linear heptapeptide, that we named subarmigeride A (1).”
Furthermore, as recommended, one sentence was added in the conclusion section, in order to present the future strategy in studying these linear peptides, (Lines 360-362) “Furthermore, purification of subarmigeride A and its analogues are in progress in order to obtain pure linear peptides, that could be evaluated for biological and environmental activities.”
We hope meet with your approval.
Looking forward to your reply,
Sincerely yours,
Dr. Marie-lise Bourguet-Kondracki
The respond to the reviewer’s new comments, written in blue color, are as follows:

Round 2
Reviewer 1 Report
accept